# Fiber Bragg Grating Sensor Networks Enhance the In Situ Real-Time Monitoring Capabilities of MLI Thermal Blankets for Space Applications

**DOI:** 10.3390/mi14050926

**Published:** 2023-04-25

**Authors:** Alessandro Aimasso, Carlo Giovanni Ferro, Matteo Bertone, Matteo D. L. Dalla Vedova, Paolo Maggiore

**Affiliations:** Department of Mechanical and Aerospace Engineering, Politecnico di Torino, 10129 Turin, Italy

**Keywords:** aerospace, Fiber Bragg Gratings, thermal measurements, thermal control, MLI, onboard system, optical fibers, sensor network, smart systems

## Abstract

The utilization of Fiber Bragg Grating (FBG) sensors in innovative optical sensor networks has displayed remarkable potential in providing precise and dependable thermal measurements in hostile environments on Earth. Multi-Layer Insulation (MLI) blankets serve as critical components of spacecraft and are employed to regulate the temperature of sensitive components by reflecting or absorbing thermal radiation. To enable accurate and continuous monitoring of temperature along the length of the insulative barrier without compromising its flexibility and low weight, FBG sensors can be embedded within the thermal blanket, thereby enabling distributed temperature sensing. This capability can aid in optimizing the thermal regulation of the spacecraft and ensuring the reliable and safe operation of vital components. Furthermore, FBG sensors offer sev eral advantages over traditional temperature sensors, including high sensitivity, immunity to electromagnetic interference, and the ability to operate in harsh environments. These properties make FBG sensors an excellent option for thermal blankets in space applications, where precise temperature regulation is crucial for mission success. Nevertheless, the calibration of temperature sensors in vacuum conditions poses a significant challenge due to the lack of an appropriate calibration reference. Therefore, this paper aimed to investigate innovative solutions for calibrating temperature sensors in vacuum conditions. The proposed solutions have the potential to enhance the accuracy and reliability of temperature measurements in space applications, which can enable engineers to develop more resilient and dependable spacecraft systems.

## 1. Introduction

In space system engineering, the thermal load is one of the fundamental aspects to be taken into account during the design and testing phase [1]. The operating temperature can vary from a few Kelvin to several hundred Kelvin according to the exposure to solar radiation. It is then mandatory to have an efficient thermal control system that deactivates components when the temperature reaches high peaks and activate cooling or warming systems if they are present onboard. For this reason, sensors that can guarantee high performances even working in harsh conditions (i.e., high temperature, electromagnetic radiation, etc.) comprise an important strategy in space applications [2,3,4,5].

Optical fiber refers to a cylindrical glass material that is capable of transmitting light through its core. Its utilization has proliferated at a significant pace, finding widespread applications across several industrial domains such as telecommunications, medical diagnostics, lighting, and the Internet, among others. As a result of its versatile functionality and wide-scale applicability, optical fiber has emerged as a pivotal technology that is extensively deployed in daily life and has permeated the global economic landscape.

Due to the extensive capabilities presented by optical fiber, Fiber Bragg Gratings (FBGs) are well-suited for the measurement of a wide range of technical characteristics in both static and dynamic modes. These sensors hold potential to replace numerous conventional sensors in aerospace applications [6,7,8], including structural monitoring, temperature regulation, and compensation. FBG sensors have already been integrated into various space systems, primarily for temperature measurement, vibration analysis, and vacuum testing for the thermal characterization of specific components. For instance, in a recent space application, FBGs were utilized to regulate the temperature of a propulsion tank owing to the fiber’s resistance to electromagnetic radiation and electrical inactivity. Furthermore, the European Space Agency’s (ESA) mission, Probe-2, employed FBG sensors for in-orbit thermal testing, while other studies have employed optical technology to detect the temperature of specific space systems [9,10].

The integration of Fiber Bragg Grating (FBG) sensors into Multi-Layer Insulation (MLI) blankets for space use offers numerous advantages in terms of monitoring and controlling the environmental conditions of spacecraft [6,7,8,9,10]. The use of FBG sensors in MLI blankets [11,12,13,14,15] enables the measurement of key parameters such as pressure, integrity, and temperature, which are critical for ensuring the proper functioning of spacecraft systems [16,17,18].

Structural health monitoring is another important application of FBG sensors in MLI blankets. By monitoring the strain and deformation of the spacecraft’s cover, FBG sensors can detect and diagnose potential failures or impacts, allowing for timely intervention and repair [19]. This can help to improve the overall reliability and safety of spacecraft, especially in the harsh and unpredictable space environment [15,20]. FBG sensors can provide pointwise temperature evaluation in MLI blankets. This is crucial for optimizing the thermal management of spacecraft systems, as temperature fluctuations can affect the performance and longevity of various components. By monitoring temperature variations at specific points in the MLI blanket, FBG sensors can enable precise control (closed-loop), deactivating the heating system or activating the radiation mechanism of the spacecraft’s thermal environment.

Finally, FBG sensors in MLI blankets can also serve as structural health sensors against hypervelocity impacts [21,22]. The harsh environment of space is fraught with hazards such as micrometeoroids and space debris, which can cause serious damage to spacecraft. FBG sensors can detect and analyze the impact of such objects, providing critical information for designing more robust and resilient spacecraft structures. The integration of FBG sensors into MLI blankets (as reported in Figure 1) offers numerous advantages for space use, including pressure and temperature monitoring, structural health monitoring, pointwise temperature monitoring, and structural health sensing against hypervelocity impacts. These benefits can help to enhance the safety, reliability, and performance of spacecraft, making them more effective in achieving their scientific and exploratory missions. The presented application of modern in situ monitoring techniques embedded into structures could provide future benefits also in Earth infrastructure monitoring [23,24,25,26].

## 2. Materials and Methods

Optical fiber is composed of multiple concentric layers: the core, cladding, and coating [27]. The core is the innermost layer and enables the transmission of light signals containing vital information. Typically manufactured from glass or polymeric materials, the core has a thickness not exceeding 50 μm. The intermediate layer, the cladding, is crucial to ensuring proper fiber operation and has a diameter of 125 μm. The coating is the outermost layer, which serves to safeguard the structure from potential damage resulting from the fiber’s low bending resistance. To enhance the mechanical strength, multiple additional outer layers may be incorporated due to the fiber’s high brittleness. An artistic scheme of the optical’s fibres is reported on Figure 2.

The sensors employed in this work were Fiber Bragg Gratings (FBGs). They were created in the fiber itself by employing a laser method to create a periodic modulation in the core’s refractive index. At the conclusion of this process, with a fiber of about 1 cm, there were some core bands with a new refractive index, resulting in $n_{f}=n_{i}+\Delta n$. Each of the parties with the changed refractive index was separated by a certain distance, denoted by the grating period *Λ_G_*. This mechanism allowed the sensor to work as a filter: when light passes through it, the FBG reflects a certain wavelength, known as the Bragg frequency, according to:(1)$$\lambda_{B}=2n_{EFF}\Lambda_{G}$$
where $\lambda_{B}$ is the wavelength reflected by the FBG, $n_{EFF}$ is the refractive index of the fiber (after the remodulation), and $\Lambda_{G}$ is the *pitch* of the grating, as shown in Figure 3. The Bragg frequency represents the output of the FBG sensor. The dependency of the Bragg frequency on the grating pitch, which is a physical distance, is shown in (1): this indicates that the fluctuation in the reflected wavelength is always related to a mechanical strain generated on the grating period by an external component.

As a result, it is simple to understand that loads applied to the sensor (in terms of induced strain) or thermal excursion create a significant variation in the reflected wavelength of the FBG, and so, (1) might be expressed as follows:(2)$$∆\lambda_{B}=K_{\varepsilon }∆\varepsilon +K_{T}∆T$$

In this way, the reflected wavelength is directly proportional to the strain and temperature variation applied to the sensor: the above-mentioned relation is then crucial in the process of sensor calibration conducted in the current study.

For the experimental test campaign, the subsequent hardware material was used: a data acquisition system composed of optical fibers and FBGs, a laser FBG interrogator, electronic temperature sensors (SHT85 or thermocouples), structural supports, and a thermo-vacuum chamber.

The FBG interrogator is a component that can automatically identify and interrogate FBGs present in fibers connected to various channels, while simultaneously collecting and analyzing their responses. The interrogator communicates independently with each sensor, reducing the likelihood of data misinterpretations arising from multiple FBGs. It transmits a laser beam through the fiber and detects the reflected wavelengths. For this application, a SmartScan SBI laser interrogator developed by the Smart Fibres company (Bracknell RG12 9BG, UK) was utilized, and is visually reported on Figure 4. The system executes a data acquisition loop once every minute, with each loop lasting one second and sampling at a variable frequency between 2.5 and 25 kHz. The average of all the data obtained for a particular measurement on a given fiber Bragg grating is computed to produce the associated instantaneous wavelength value. The data are transmitted to the PC through a LAN connection. A schematic picture of the acquisition system is reported on Figure 5.

In the evaluation of the FBGs’ performance, thermocouples were employed in the thermo-vacuum chambers. Thermocouples are temperature transducers that operate based on the Seebeck effect. The thermo-vacuum chamber is utilized to recreate the environmental conditions present in space. A picture of the chamber used in the experiments is reported on Figure 6. By concurrently controlling the pressure and temperature within a well-defined volume of space, the only modes of heat transfer are conduction and radiation, just as in space. The chamber can depressurize the test environment to values as low as 1 × 10^−8^ mbar, while operating within temperature ranges of −190 °C to +160 °C, and the minimum temperature value is dictated by the use of nitrogen. These temperature and pressure ranges may be altered based on the thermal and pumping capabilities of the vacuum chamber. For instance, the temperature ranges can be expanded using heating lamps (IR or solar simulation) and/or cryo-coolers, which utilize helium thermodynamics, enabling temperatures similar to deep space to be reached theoretically.

For the test campaign, four fibers with polyimide coating were selected. The polyimide coating was selected as the only material, which is easily available on the market, able to satisfy the thermal and outgassing requirements necessary to carry out the tests in the thermo-vacuum chamber. The fibers were affixed to the metal surface exclusively using a simple adhesive film, requiring no further preparation before testing. A positioning detail picture and an experimental setup in thermos-vacuum chamber are reported on Figure 7.

Several preliminary measurement cycles were conducted in a climate chamber to determine the optimal data acquisition technique, in order to minimize unwanted effects and errors.

The evaluation of the FBG’s performance encompassed multiple stages, comprising:Acquiring raw data.Calculating the *λ*(*T*) curve.Converting the FBG output into temperature values.Determining the error.

Initially, raw data were collected by plotting the information stored in the .log files generated by the interrogator using MATLAB©. Subsequently, the sensor’s characteristic *λ*(*T*) relation was established from the raw data, elucidating the relationship between increasing temperature and sensor output, independent of the chronological time history of the setup temperature.

The *λ*(*T*) relation is explained in terms of:(3)$$\lambda \left(T\right)=\lambda_{0}+K_{T}T$$
where *K_T_* and *λ*_0_ are the angular coefficients and the known term of the linear fit calculated from the experimental data.

Moreover, considering that sensors have different nominal Bragg wavelengths, the relation could be normalized as follows:(4)$$\frac{\Delta \lambda }{\lambda }=k_{0}+k_{T}T$$

From this calibration, it was possible to convert the FBG reflected wavelength into a temperature value using the equation:(5)$$T\left(\lambda \right)=\frac{\lambda -\lambda_{0}}{K_{T}}$$

To generalize the relation, the temperature could be calculated from the normalized relation as follow:(6)$$T\left(\lambda \right)=\frac{\left(\frac{\Delta \lambda }{\lambda }\right)-k_{0}}{k_{T}}$$

After the preliminary analysis, the FBG sensor network was placed in the thermo-vacuum chamber for measurements across a broad temperature range of approximately −150 °C to 200 °C, based on the outcomes of the laboratory tests conducted in the climate chamber. The sensors were mounted on a Kapton [30] thermal blanket, which is typically employed in space applications, such as thermal blankets for thermal control systems. Specifically, no tension was exerted on the fibers, and no adhesive was placed near the sensors, reproducing the free fiber conditions observed in the initial climate chamber test.

The same aforementioned procedures were followed during this test, encompassing:Collecting raw data.Calculating the *λ*(*T*) curve.Converting the FBG output into temperature values.Determining the error.

The equations employed to convert the raw data into temperature values were identical to those described previously in this section. Since the objective was to utilize FBGs for dependable monitoring of space components, the thermal cycles carried out here were more intricate. Specifically, the experiment comprised distinct steps of approximately 15 min at a stable temperature, followed by a transitional phase to attain a new stable step. The test campaign was divided into three sections:Thermal Cycle 1 ranging from 0 °C to 200 °C.Thermal Cycle 2 ranging from −150 °C to 200 °C.Repeatability and accuracy validation cycle.

During the first two experiment sessions, each FBG was paired with a thermocouple, and the calibration coefficients were determined. In the final session, the thermal cycle from the second session was repeated to verify the accuracy of the FBG response. As specified before, optical fibers with a polyimide coating were utilized to withstand the high thermal excursion. A simplifying flow chart of the overall activity described is disclosed on Figure 8.

## 3. Results

The findings derived from the test campaigns are expounded hereafter, based on the two distinct thermal cycles and a repeatability analysis, as outlined in the preceding Section 2.

### 3.1. Thermal Cycle 1: Temperature Range 0 to 200 °C

Initially, the sensors underwent a testing phase where they were subjected to a temperature range of 0 °C to 200 °C. The objective of this testing was to examine the linear relationship between the temperature and the output of the Fiber Bragg Grating (FBG). The results demonstrated that the *T*(*λ*) characteristic exhibited a linear correlation between the temperature and the variation of the FBG output. Additionally, the coefficient *K_T_* was found to be similar to that detected in previous tests performed in a climate chamber.

This finding suggested that the FBG outcomes can be applied in a vacuum environment in the same manner as in atmospheric conditions, making them useful for both aeronautical and space applications. Furthermore, the errors detected after converting the sensors’ outputs into temperature data were minimal, and the vacuum environment effectively eliminated the mechanical disturbances caused by convective motion that were previously present in non-vacuum measurements [31]. The experimental outcomes derived from the first session of experiments are reported on Figure 9.

Ultimately, with regard to the bonding technique, it was imperative to choose materials with thermally stable properties, even in a vacuum environment. The application of adhesive directly onto the sensors was avoided to prevent the generation of mechanical stresses. The error trends and the boxplot of the *K*(*T*) are reported on Figure 10.

### 3.2. Tests Cycle 2: Temperature Range −150 to 200 °C

Building upon the positive outcomes obtained from the initial segment of the experiment, the entire optical sensor network was subjected to negative temperatures, reaching −150 °C, to investigate potential non-linear phenomena in the calibration curve [32,33,34]. The principal outcome of this test was that all the Fiber Bragg Grating (FBG) sensors demonstrated an analogous *λ*(*T*) calibration curve trend (Figure 10a). Specifically, the curve exhibited essentially linear behavior up to a transition point of approximately −50 °C. Subsequently, the slope of the calibration curve *λ*(*T*) reduced for all sensors by the same amount. The sole disruption detected was a minor delay in the response time of the FBG in comparison to the thermocouple for the points located in close proximity to the heater. The consistent trend facilitated the approximation of the *λ*(*T*) curve with a linear stroke. In particular, one *K_T_* coefficient was employed for temperatures above −50 °C, and a second, lower coefficient was employed for temperatures below this threshold. The linear stroke approximation already demonstrated a high level of accuracy and dependability. Nevertheless, the accuracy of the calibration process was further refined by applying numerical approximation methodologies, as illustrated in Figure 11. In conclusion, the observations made regarding positive temperatures can be extended to low temperatures by altering the *K_T_* coefficient below −50 °C for the present configuration.

### 3.3. Repeatability and Accuracy Analysis

The reliability of the Fiber Bragg Grating (FBG) measurements is contingent on their repeatability, making it a crucial requirement for thermal testing. Therefore, a final test was conducted to verify the previously obtained thermal characterization. In this experiment, all the thermocouples, except one, were removed, and the overall thermal cycles were repeated.

To ensure consistency, the chamber was stabilized at the same temperatures, and the temperature was calculated by employing the calibration coefficients derived from the previous campaign. The raw data obtained from both tests were initially compared, and it was observed that the FBG recorded stabilization on the same previous wavelength when the chamber imposed the same temperature.

Figure 12 depicts the perfect coherence between the two tests, with the only differences being the duration of the steps and the final return to environmental conditions after the lower step, due to the manually controlled process.

The final outcome pertained to the level of accuracy achieved in the temperature reading following stabilization. As shown in Figure 13, an accuracy capable of detecting a stability of less than 0.2 °C/h, a typical requirement in the space industry, was attained. Furthermore, the moving average value of the fiber over a 60 min period is plotted with a tolerance band of ±0.1 °C (i.e., 0.2 °C/h amplitude). The graph illustrates that the oscillations remained within the tolerance band. As demonstrated in the graph on the right, the error committed with respect to the anticipated mean value was less than 0.1 °C/h in absolute terms, which is significantly below the required constraint.

## 4. Discussion

The experimental campaign conducted in this study yielded highly favorable and promising results, indicating that Fiber Bragg Grating (FBG) sensors can be exceptionally useful for the thermal characterization of components in space applications. These sensors possess high sensitivity, enabling them to detect even minor temperature variations, and they are smaller in size compared to traditional sensors such as thermocouples, allowing a low weight and flexibility to the MLI, ideal for deployable or inflatable structures. Additionally, FBGs exhibit a shorter response time and can instantaneously detect sudden thermal changes, unlike thermocouples. Moreover, the presence of multiple Bragg sensors on a single optical fiber allows for precise information at numerous points with only one cable, whereas electronic thermocouple sensors necessitate one device per point, resulting in greater interference.

Consequently, the measurement cycle was conducted in a vacuum, using a free fiber solely affixed to the specimens at temperatures typical of the space environment. This approach facilitated the integration of the instrument onto the tested supports, such as metal plates and/or thermal protection coatings, without compromising measurement accuracy. Under stable mechanical conditions, the measurement cycles demonstrated the high reliability of the outputs and the complete elimination of interferences and noise in the data. The ease of the integration strategy adopted in this study has significant implications for potential industrial applications in the future.

The final measurement campaigns revealed that FBGs can be safely used at operating temperatures of up to 200 °C, a value that is seldom supported by traditional electronic sensors, other than thermocouples. However, special attention must be paid to negative temperatures. In all the tests conducted, the linear characteristic of *T*(*λ*) underwent significant changes. During the tests, a critical temperature of −50 °C was identified, below which a new linear fit could be obtained from the experimental data. Nevertheless, this more complex calibration enabled accurate detection of low temperatures by FBGs.

The last segment of the tests demonstrated the ability of FBGs to autonomously detect temperature, with a remarkably high level of accuracy.

## 5. Conclusions

The favorable outcomes of the comprehensive test campaigns suggested that Fiber Bragg Grating (FBG) sensors have considerable potential for space applications, especially for thermal characterizations, owing to the substantial number of sensors available and the extremely compact size of the cable. Additionally, the FBGs’ high sensitivity enabled them to detect not only temperature variations, but also other measurements such as strain or damage due to micrometeoroid impact.

Further investigations are necessary to examine the behavior of FBGs in cryogenic cases and to gain a better understanding of the transition phase between the two linear fits determined by the experimental data.

Finally, this study emphasized the importance of establishing precise standards for developing specific sensor packaging. This is a challenging task due to the fiber’s exceptional sensitivity to the environmental conditions, including the temperature and mechanical conditions.

Overall, the potential of FBG sensors for space applications, particularly in thermal characterizations, offers a promising avenue for advancing technology in the field, with further research and development poised to yield even more impressive results.

## Figures and Tables

**Figure 1 micromachines-14-00926-f001:**
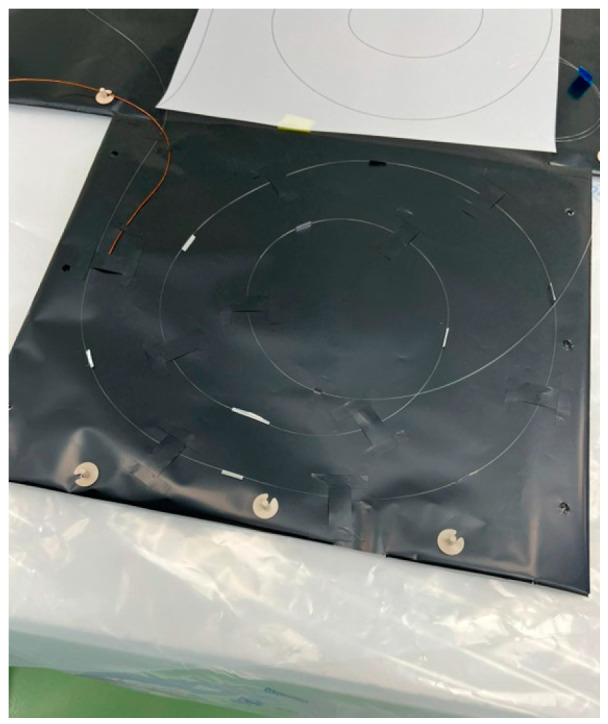
MLI blankets with integrated FBG sensors.

**Figure 2 micromachines-14-00926-f002:**
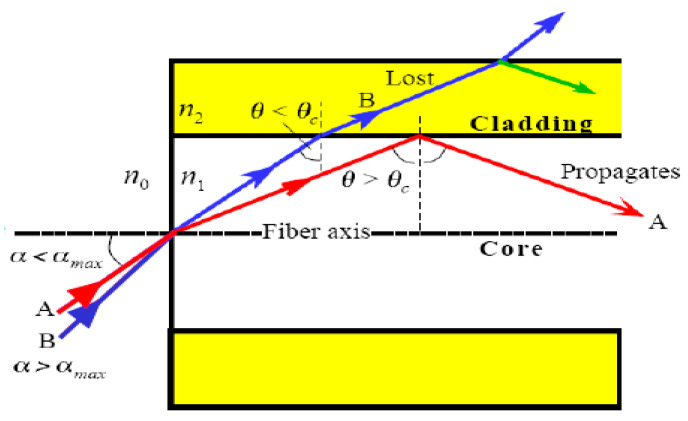
Scheme of the optical fibers’ working principle [28].

**Figure 3 micromachines-14-00926-f003:**
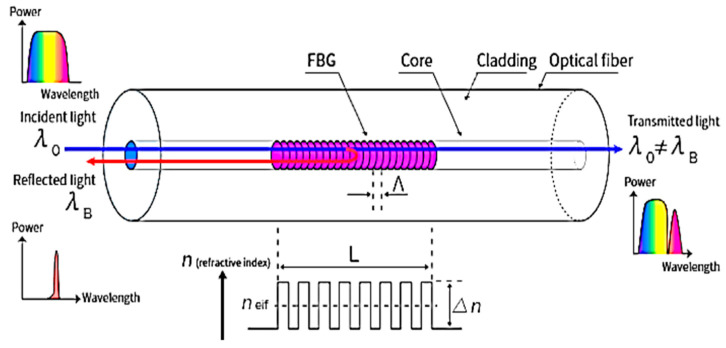
Scheme of FBG working principle [29].

**Figure 4 micromachines-14-00926-f004:**
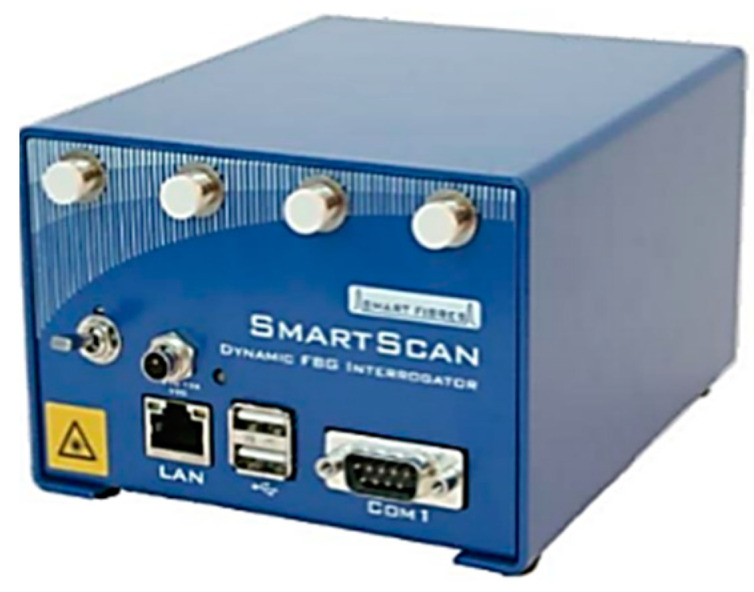
The Smart Scan interrogator used in the tests.

**Figure 5 micromachines-14-00926-f005:**
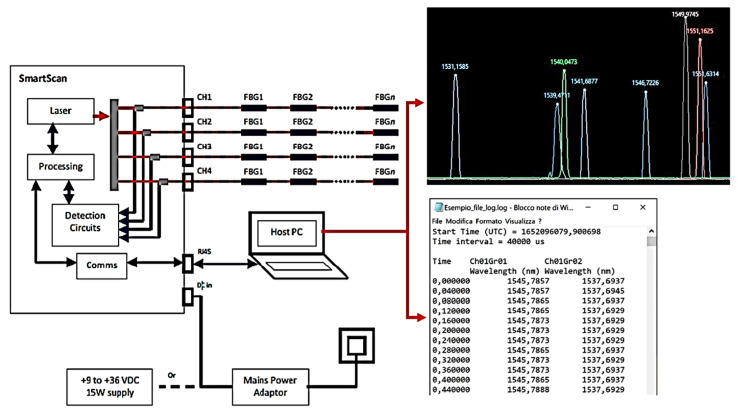
Scheme of the data acquisition system.

**Figure 6 micromachines-14-00926-f006:**
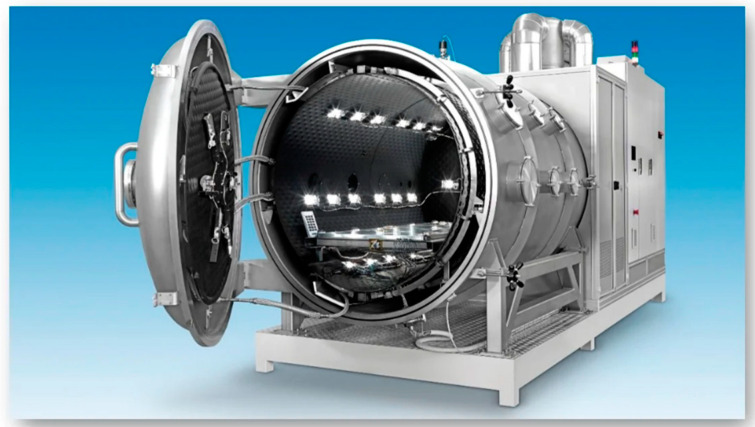
The thermo-vacuum chamber employed in the tests.

**Figure 7 micromachines-14-00926-f007:**
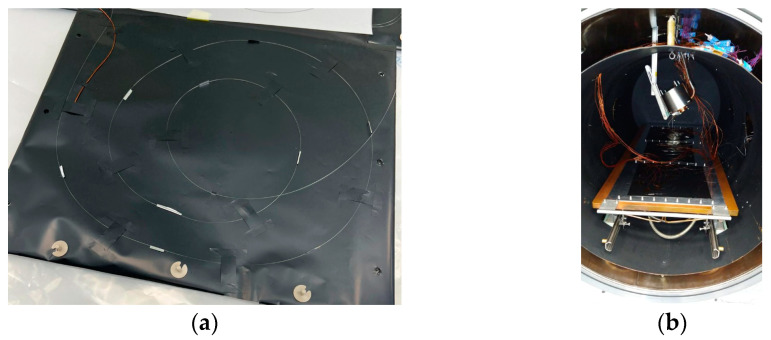
FBG positioning on Kapton support (**a**) and experimental setup in vacuum (**b**).

**Figure 8 micromachines-14-00926-f008:**
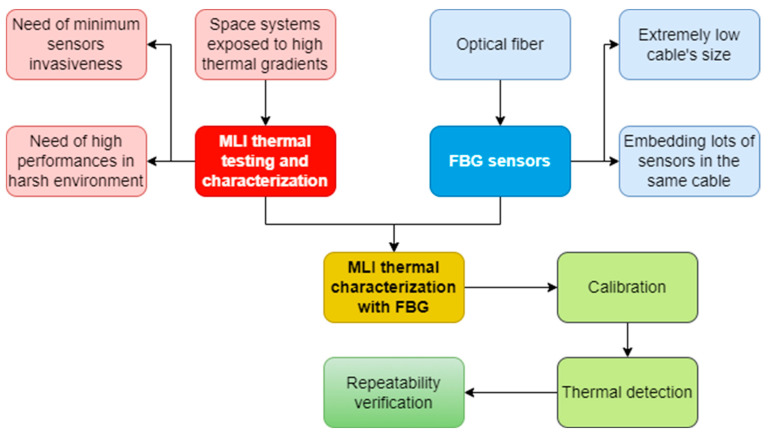
Flowchart of the overall activity described in the work.

**Figure 9 micromachines-14-00926-f009:**
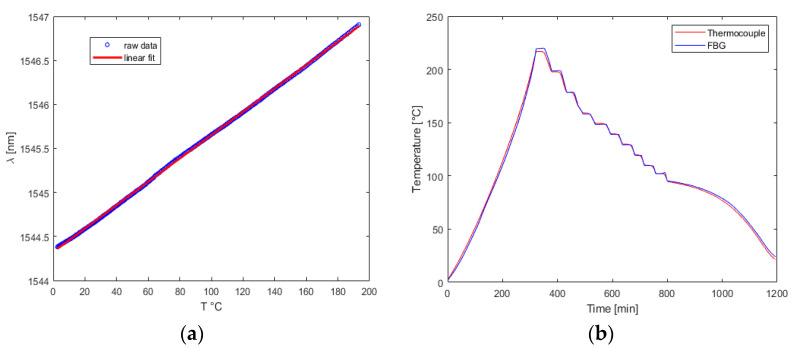
Test Cycle 1: (**a**) single FBG calibration curve; (**b**) comparison between FBG and thermocouple.

**Figure 10 micromachines-14-00926-f010:**
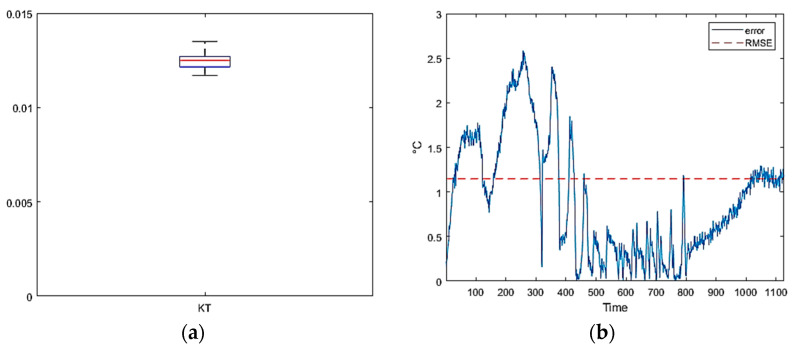
Test Cycle 1: (**a**) boxplot of experimental coefficient; (**b**) absolute error trend.

**Figure 11 micromachines-14-00926-f011:**
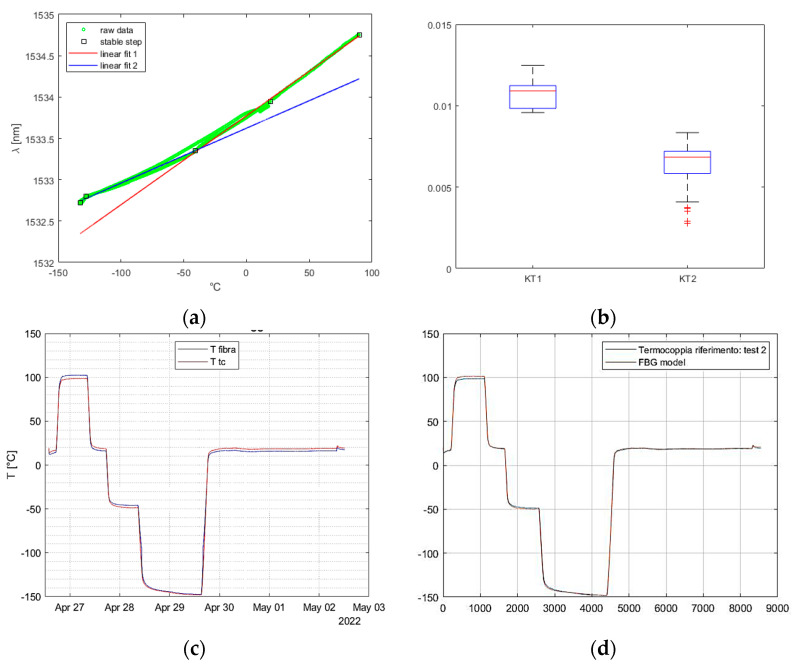
Test Cycle 2: (**a**) single FBG calibration curve; (**b**) boxplot of calculated coefficients; (**c**) linear interpolated data; (**d**) Fourier interpolated data.

**Figure 12 micromachines-14-00926-f012:**
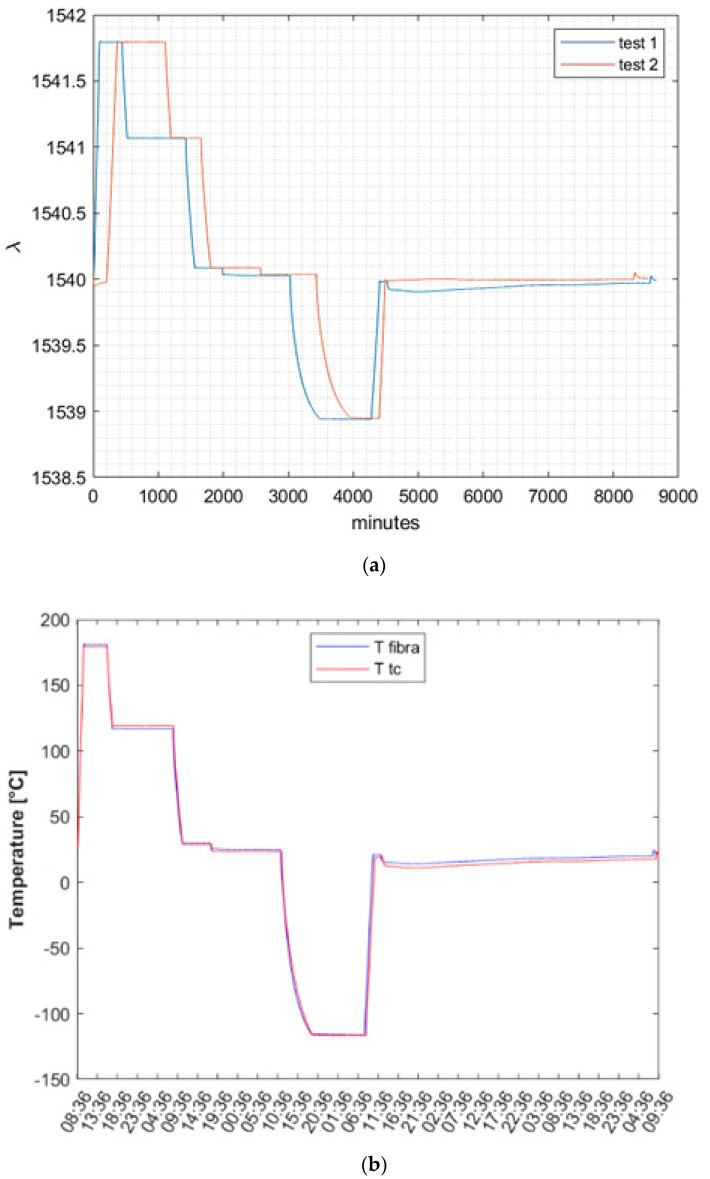
Repeatability data analysis: (**a**) comparison between row data of the same FBG in the two thermal cycles; (**b**) comparison between thermocouple (red) and FBG (blue) measurements with the FBG already calibrated in the previous test.

**Figure 13 micromachines-14-00926-f013:**
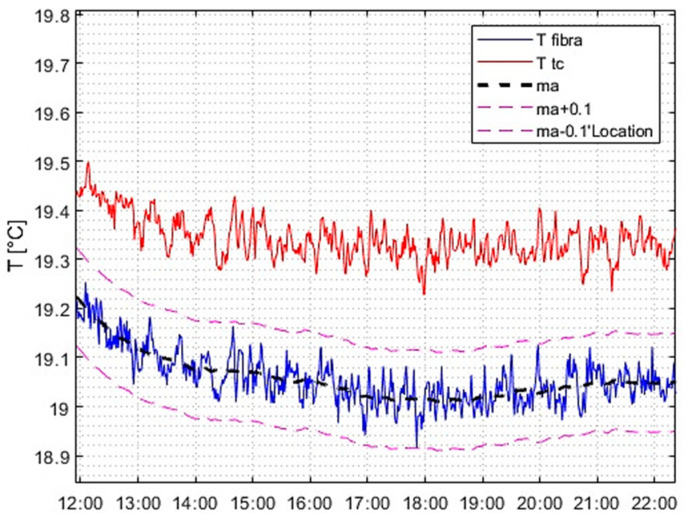
Detailed comparison between FBG and thermocouple performances and stability.

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
