# Peer review of "Fiber Bragg Grating Sensor Networks Enhance the In Situ Real-Time Monitoring Capabilities of MLI Thermal Blankets for Space Applications"

_micromachines, 2023, doi:10.3390/mi14050926_

Round 1
Reviewer 1 Report
1. Since the purpose of the article is to investigate innovative solutions for calibrating temperature sensors in vacuum conditions,could the author provide a flow chart of this solution so that readers can understand it more clearly.
2. I think Fig.7 (a) Fig.1 are the same, but of different dimensions. Please explain why.
3. Please explain why polyimide coating is used instead of other materials.
4. Some recent relevant references should be included, such as “YM Zhang, ZM Xiao, M. Fan. Fatigue investigation on railway wheel steel with white etching layer. International journal of steel structures (2020), 20: 80-88.” “Yuan XW, Xiao ZM and Zhang YM. Numerical investigation of fracture behavior by crazing in graphene reinforced polymers. Mechanics of Advanced Materials and Structures 2021; 29: 7703-7711. DOI: 10.1080/15376494.2021.2005191” “YM Zhang, ZM Xiao and J Luo. Fatigue crack growth investigation on offshore pipelines with three-dimensional interacting cracks. Geoscience Frontiers (2018), 9 (6): 1689-1697.” “Yuan XW, Li WG, Xiao ZM, Zhang YM. Prediction of temperature-dependent transverse strength of carbon fiber reinforced polymer composites by a modified cohesive zone model. Composite Structures 2023; 304. DOI: 10.1016/j.compstruct.2022.116310”.
The language use is good.
Author Response
- Since the purpose of the article is to investigate innovative solutions for calibrating temperature sensors in vacuum conditions,could the author provide a flow chart of this solution so that readers can understand it more clearly.
The following scheme has been added to Materials and Method section.
- I think Fig.7 (a) Fig.1 are the same, but of different dimensions. Please explain why.
In fact the two figure are of the same object with different zoom levels. The first one presents a general layout of a MLI while the second, more detailed and more closed intend to evidence the positioning of the FBG on the Kapton support.
- Please explain why polyimide coating is used instead of other materials.
The polyimide coating was selected as the only material, from them easily available on the market, able to satisfy the thermal and outgassing requirements necessary to carry out the tests in the thermo-vacuum chamber
- Some recent relevant references should be included, such as:
“YM Zhang, ZM Xiao, M. Fan. Fatigue investigation on railway wheel steel with white etching layer. International journal of steel structures (2020),
“Yuan XW, Xiao ZM and Zhang YM. Numerical investigation of fracture behavior by crazing in graphene reinforced polymers. Mechanics of Advanced Materials and Structures 2021; 29: 7703-7711. DOI: 10.1080/15376494.2021.2005191”,
“YM Zhang, ZM Xiao and J Luo. Fatigue crack growth investigation on offshore pipelines with three-dimensional interacting cracks. Geoscience Frontiers (2018), 9 (6): 1689-1697.,
“Yuan XW, Li WG, Xiao ZM, Zhang YM. Prediction of temperature-dependent transverse strength of carbon fiber reinforced polymer composites by a modified cohesive zone model. Composite Structures 2023; 304. DOI: 10.1016/j.compstruct.2022.116310”.
References added.

Reviewer 2 Report
A good work that deserves publication.
The research presented in this manuscript is intended to an audience related to space technology so some lighter treatment of the FBG sensors is acceptable.
The minor revision that I suggest is related to some minor language issues (next separator) and the initial part of Materials and Methods.
The basic initial treatment of optical fibers with Snell law (1) and the input maximum angle (2) is so away of the focus of the paper that could be removed (the superficial treatment of FBG is OK)
I found a few words that don’t sound well in the text. I suggest to try to find a replacement:
Line 125- thermal excursion
Line 152- is utilized to
Line 165- were affixed to
Author Response
Following your review, the introduction about the optical fiber working principle has been resumed by removing the equations of Snell law and the input maximum angle
